# Structural determinants of dynamic fluctuations between segregation and integration on the human connectome

Makoto Fukushima [1,2,3 ✉] & Olaf Sporns [4,5]

While segregation and integration of neural information in the neocortex are thought to be important for human behavior and cognition, the neural substrates enabling their dynamic fluctuations remain elusive. To tackle this problem, we aim to identify specific network features of the connectome that are responsible for the emergence of dynamic fluctuations between segregated and integrated patterns in human resting-state functional connectivity. Here we examine the contributions of network features to dynamic fluctuations by constructing rewired surrogate connectome in which network features of interest are selectively preserved, and then by assessing the magnitude of fluctuations simulated with these surrogates. Our analysis demonstrates significant contributions from global geometry and topology of the connectome, as well as from localized structural connections involving visual areas. By providing structural accounts of dynamic fluctuations in functional connectivity, this study offers new insights into generative mechanisms driving temporal changes in segregation and integration in the brain.

[1] Division of Information Science, Graduate School of Science and Technology, Nara Institute of Science and Technology, 8916-5 Takayama-cho, Ikoma, Nara 630-0192, Japan. [2] Data Science Center, Nara Institute of Science and Technology, 8916-5 Takayama-cho, Ikoma, Nara 630-0192, Japan. [3] Center for Information and Neural Networks, National Institute of Information and Communications Technology, 1-4 Yamadaoka, Suita, Osaka 565-0871, Japan. [4] Department of Psychological and Brain Sciences, Indiana University, 1101 East 10th Street, Bloomington, IN 47405, USA. [5] Indiana University Network Science Institute, 1001 IN-45, Bloomington, IN 47408, USA. ✉email: mfukushi@is.naist.jp

Advances in measuring techniques of white matter structural connectivity allow obtaining whole-brain network maps, referred to as the connectome[1]. Complex patterns of structural connections have been investigated using methods widely adopted in network science research[2]. Notable network features of the connectome are its community structure and its interconnected hubs. Brain regions within communities are densely connected to each other[3–5], supporting local communication within functionally segregated systems in the brain. In parallel, some brain regions connect with many others across diverse communities. These hub regions form a densely interconnected network core[6–8] (but also Rubinov[9]), promoting global integration of information processed within segregated communities.

Through communities and interconnected hubs, the connectome provides a structural backbone on which functional segregation and integration of neural information[10] can fluctuate spontaneously or in response to momentary demands from the environment[11,12]. The balance between segregation and integration is essential for balancing effective local processing and global communication of neural information, which jointly support cognition. Recent functional magnetic resonance imaging (fMRI) studies suggest that this balance manifests in changing patterns of co-activations (i.e. functional connectivity) among brain regions[13,14]. For instance, Cohen and D'Esposito[13] demonstrated that segregated and integrated patterns of functional connectivity were critical for motor execution and working memory tasks, respectively, and that flexible reorganizations between these two patterns were related to better behavioral performance. While Cohen and D'Esposito[13] used functional connectivity computed from the entire fMRI scan durations, Shine et al.[14] used functional connectivity on a time scale of tens of seconds (for a review of this time-resolved functional connectivity, see Preti et al.[15]). Dynamic fluctuations of fMRI time-resolved functional connectivity were first reported in humans[16], and later in other species[17,18]. While there is ongoing discussion regarding the stationarity of fMRI functional connectivity[19,20], an emerging consensus affirms the existence of state- and task-related fluctuations on functional networks and favors the use of rigorous statistical tests and models for their robust estimation[21]. A prominent feature of time-resolved functional connectivity is ongoing fluctuation between more segregated and more integrated connectivity patterns[14,22–25]. Shine et al.[14] demonstrated that fluctuations between segregated and integrated time-resolved connectivity patterns shifted toward more integration with greater task demands and that this shift also supported fast and effective performance on working memory tasks. Furthermore, fluctuations between segregated and integrated patterns during rest were correlated with fluctuations in pupil diameter, an index of state-dependent neuromodulatory activity[14]. The fluctuating dynamics between segregation and integration are thus related to a variety of aspects of human cognition and behavior.

Despite these observations, the specific neural substrates enabling dynamic fluctuations between segregated and integrated connectivity patterns remain largely unknown. In a recent study[26], Shine and colleagues applied a computational method to investigate the origin of dynamic fluctuations between segregation and integration. They simulated resting-state fMRI (rs-fMRI) data using local models of oscillatory neural dynamics that are globally coupled based on the connectome[27–29]. They found that externally controlling a neural gain parameter in the simulation model can alternate connectivity patterns between segregation and integration. This finding suggests that active processes of neural gain control by neuromodulatory systems may contribute to the appearance of fluctuating connectivity patterns. In parallel, rs-fMRI data were simulated using neural oscillator models

coupled based on the connectome, and the extent to which this simulated data can replicate dynamic fluctuations between segregation and integration in empirical rs-fMRI data was investigated without an explicit control of the neural gain[30]. It was shown that the simulated data can reconstruct ~80% of the magnitude of empirical dynamic fluctuations in global network measures that characterize connectivity patterns of segregation and integration. This finding suggests that, in addition to potential neuromodulatory mechanisms, the intrinsic organization of the structural connectome may contribute to the emergence of dynamic fluctuations between segregation and integration.

Here we attempt to uncover which specific network features of the connectome can account for the magnitude of dynamic fluctuations between segregation and integration. First, we examined the contribution of geometry of the connectome. The connectome is a spatially embedded network whose nodes (brain regions) and edges (structural connections) are tied to physical locations. Spatial embedding entails that nodes are more likely to be connected within their spatial neighborhoods, resulting in conservation of wiring cost[9,31,32]. We evaluated the role of this geometric factor to dynamic fluctuations by simulating brain activity using surrogate connectome data in which connectivity weights were permuted while preserving the spatial relationship between weights and lengths of structural connections[33,34]. Second, we investigated the contribution of topology of the connectome, especially for the representative topological features introduced above, communities and interconnected hubs. We evaluated their potential contributions to dynamic fluctuations by using surrogate connectome data in which the weight permutation was restricted to preserve the topologies of communities and interconnected hubs. Finally, we examined the contributions of local edges in the connectome using surrogate connectome data. The purpose of this analysis was to find out which local edge sets prominently contribute to dynamic fluctuations in simulations. To identify them, we searched over sets of edges connected to each of the known resting-state networks (RSNs)[35,36].

Our surrogate data analyses demonstrated that both geometry and topology of the structural connectome significantly contributed to shaping the magnitude of dynamic fluctuations between segregated and integrated patterns of simulated functional connectivity. This magnitude was, however, not fully accounted for either by geometry, topology, or their combination. The residual amount of the magnitude in dynamic fluctuations was explained by the contribution of localized structural connections involving the visual network.

## Results

**Constructing surrogate data that preserve geometry and/or topology of the connectome.** We performed surrogate data analysis to evaluate the contributions of geometry and topology of the connectome to dynamic fluctuations between segregation and integration. The whole workflow of this analysis is summarized in Fig. 1. We first constructed surrogate connectome data that preserve geometric and/or topological features of the actual connectome data. These surrogate data were constructed by randomly permuting non-zero weight edges of the group-level structural connectome data with constraints imposed on their geometric and/or topological properties (1 in Fig. 1). The geometric constraint was used for preserving the spatial relationship between weights and lengths of structural connections in the connectome[33,34]. This constraint restricted the weight permutation within edges of similar streamline lengths (see Fig. 2a). The topological constraint consists of two parts and these were jointly used for preserving communities and interconnected hubs of the

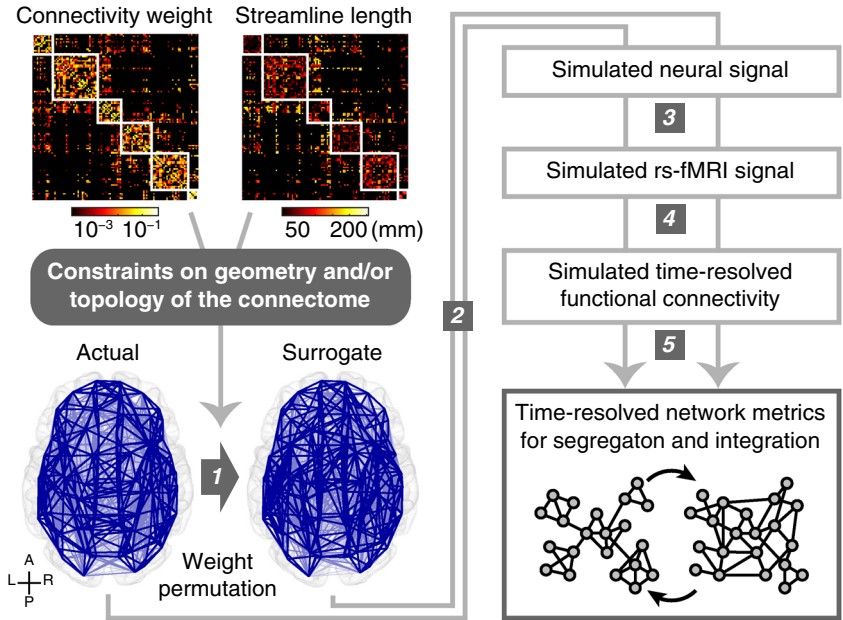

**Fig. 1 Workflow of surrogate data analysis.** (1) Constructing surrogate data of the connectome by permuting structural connectivity weights of the actual data with geometric and/or topological constraints depicted in Fig. 2. Blue lines represent regional structural connections (A: anterior; P: posterior; L: left; R: right). (2) Simulating spontaneous oscillatory neural signal using phase oscillator models coupled based on structural connectivity weights and lengths in the connectome. (3) Converting simulated neural signal into simulated rs-fMRI signal using a hemodynamic model. (4) Computing time-resolved functional connectivity of simulated rs-fMRI signal using tapered sliding windows. (5) Calculating global network measures of time-resolved functional connectivity to track dynamic fluctuations between segregated and integrated patterns of functional connectivity. Small circles represent brain regions, and lines between them indicate regional functional connections.

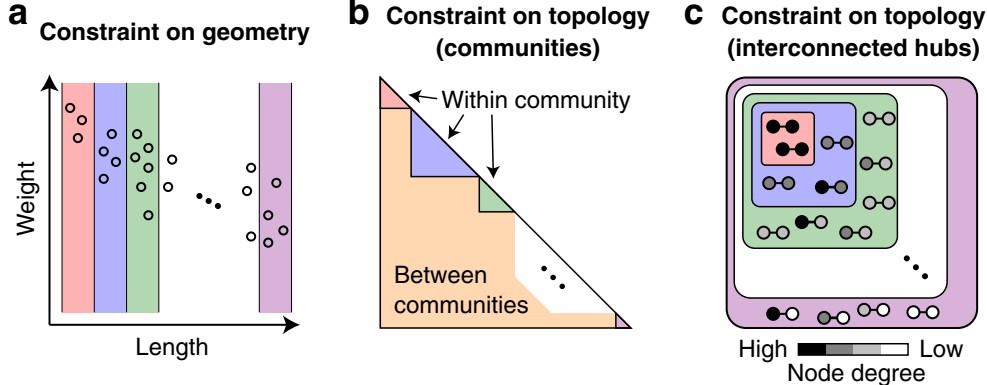

**Fig. 2 Schematics of geometric and topological constraints during the weight permutation. a** Constraint on geometry to preserve the spatial relationship between weights and lengths of structural connections in the connectome. A small circle corresponds to an edge in networks. Weights were permuted within each bin of lengths. The manner to determine the number of length bins is described in "Methods section". **b** Constraint on topology to preserve the community structure of the connectome. Weights were permuted within each community or between communities. **c** Constraint on topology to preserve the interconnected hub structure of the connectome. Small circles and lines between them represent nodes and edges, respectively. Weights were permuted within edges connecting a node of degree = $d$ and another node of degree ≥ $d$. Colors of bins in **a**, colors of communities in **b**, and colors of edge categories in **c** are independent and not related to each other.

connectome. These constraints restricted the weight permutation within edges in each community or between communities (see Fig. 2b), and within edges connecting a node of a certain degree $d$ and another node of degree ≥ $d$ (see Fig. 2c). We referred to the surrogate data without constraint as $R$, with the geometric constraint only as $G$, with the topological constraints only as $T$, and with the combined geometric and topological constraints as $GT$.

Exemplars of the surrogate data $R$, $G$, $T$, and $GT$ are presented in Fig. 3a. Spatial layout of $GT$ was very similar to that of the actual connectome data because the number of edges that can be permuted in $GT$ was less than a half of that in the other types of

the surrogate data. The fraction of the number of permuted edges averaged over surrogate samples to the number of all non-zero weight edges was 1.00, 0.98, 0.91, and 0.40 for $R$, $G$, $T$, and $GT$.

Permuting edge weights broke the original sequence of node strength (i.e. weighted degree) in the actual connectome data. To solve this problem, we applied a weight adjustment method[33,34] to preserve the strength sequence of the actual data. Applying this method after the weight permutation did not essentially affect the extent to which $G$ and $GT$ preserved geometry (see Fig. 3b) and $GT$ preserved topology of the actual connectome data (see Fig. 3c, d). In $T$, the community structure was remained preserved

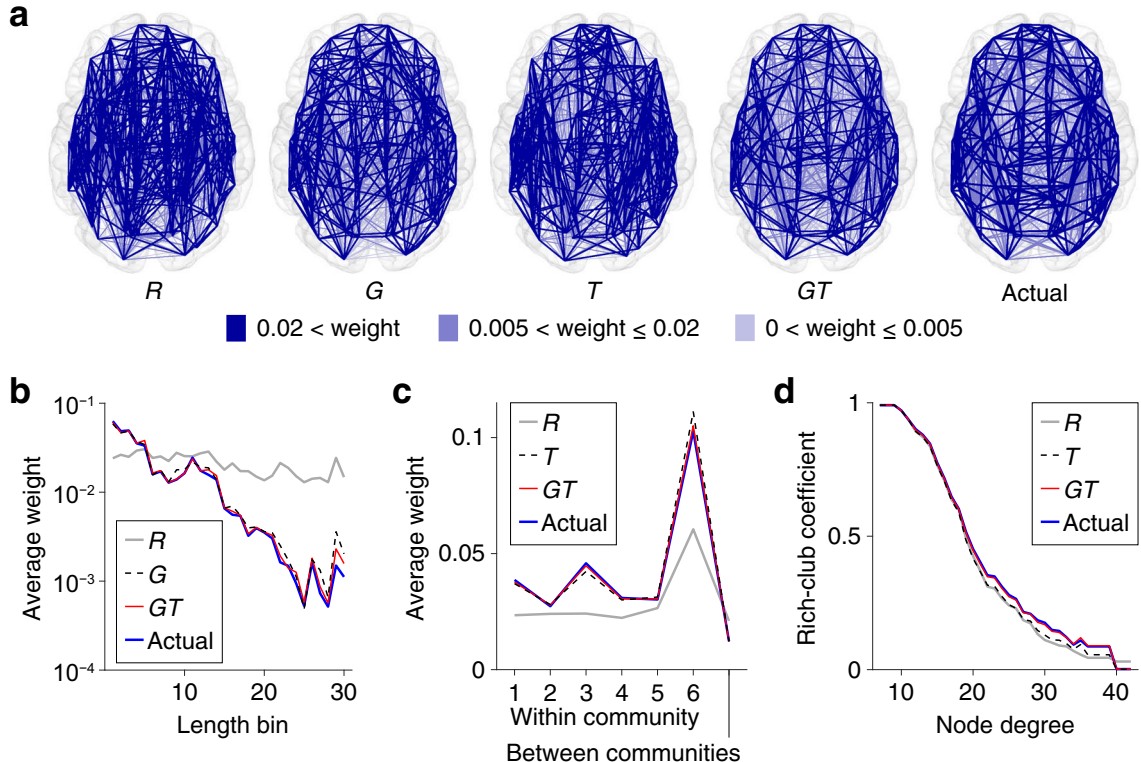

**Fig. 3 Profiles of the surrogate and actual connectome data. a** Spatial layout of the connectome. Blue lines represent regional structural connections. Exemplars are shown for the surrogate data *R*, *G*, *T*, and *GT*. Edges with smaller connectivity weights are superimposed by edges with larger connectivity weights in these graph plots. **b** Connectivity weights averaged over edges within each bin of streamline lengths. The weight–length relationship of the actual data was preserved both in *G* and *GT*. **c** Connectivity weights averaged over edges within each community or between communities. The weight profile of the actual data over the horizontal axis was preserved both in *T* and *GT*. **d** The weighted rich-club coefficient[7] (i.e. the ratio of the sum of edge weights between hub nodes to the sum of all edge weights). The coefficient profile of the actual data over the degree to define hub nodes was preserved in *GT*, but not in *T*.

(see Fig. 3c) while the interconnected hub structure became less pronounced (see Fig. 3d). Thus, it should be noted that the contribution of topology as assessed by *T* later is mainly attributable to the contribution of the community structure alone.

**Simulating dynamic fluctuations between segregation and integration on the connectome.** The contributions of connectome geometry and topology to the fluctuating dynamics of segregation and integration were examined by comparing the magnitudes of such fluctuations in rs-fMRI data simulated with the actual and surrogate connectome data (2–5 in Fig. 1). We first simulated spontaneous oscillatory neural signal using a variant of the Kuramoto model[37], consisting of simple phase oscillators coupled based on structural connectivity weights and lengths of the connectome[30,38,39]. Simulated neural signal was converted into simulated rs-fMRI signal using the Balloon/Windkessel hemodynamic model[40,41]. From simulated rs-fMRI signal, time-resolved functional connectivity was computed using tapered sliding windows[15]. For each instance of time-resolved functional connectivity, three global network measures, mean participation coefficient (mean PC), mean temporal participation coefficient (mean TPC), and modularity, were calculated to track dynamic fluctuations between segregated and integrated patterns of functional connectivity[14,23,24,42]. The time series of mean PC, mean TPC, and modularity served as proxies of dynamic fluctuations between segregation and integration in the brain. In this paper, we present results obtained from the mean PC in the main figures.

The computational model used for simulating neural signal has two free parameters, the global coupling constant *k* and the mean delay $\bar{\tau}$. In the simulations with the actual connectome data, these parameters were specified as $k = 55$ and $\bar{\tau} = 12$ ms based on systematic parameter selection under the objective of accurate reconstruction of empirical functional connectivity profiles (for details of the parameter search procedure, see "Methods section" or Fukushima and Sporns[30]). In the simulations with the surrogate data, we fixed $\bar{\tau}$ as 12 ms to save computation time, but searched for *k* in each surrogate sample for which the mean of the order parameter (i.e. the global synchrony level) of simulated neural signal most closely approximated that obtained from the actual data with $k = 55$. Depending on *k* and the type of the surrogate data, the global synchrony level greatly changed between zero (fully incoherent) and one (fully synchronized) in simulation samples generated from the connectome (see Fig. 4a). Distributions of *k* selected in *R*, *G*, *T*, and *GT* are shown in Fig. 4b.

**Both geometry and topology contribute to the emergence of dynamic fluctuations.** To assess the contributions of connectome geometry and topology to the segregation–integration dynamics, we compared the magnitudes of dynamic fluctuations in mean PC between different types of the surrogate data. Figure 4c presents distributions of these magnitudes of dynamic fluctuations, shown as the standard deviation (SD) of the simulated time series of mean PC. The magnitudes of dynamic fluctuations obtained from the geometry-constrained surrogate data *G* and the topology-constrained surrogate data *T* were both greater

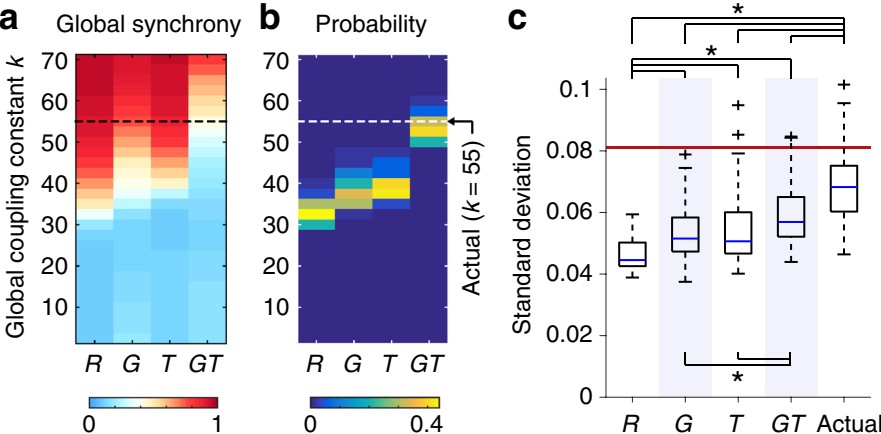

**Fig. 4 Dynamics of simulated neural signal and functional connectivity patterns. a** The global synchrony level of neural signal simulated with the surrogate data. The averages over 50 simulation samples are presented. The global synchrony level similar to those averaged over simulation samples generated from the actual data (0.37) is shown in close to white. **b** Probability distribution of the global coupling constant $k$ selected so that the global synchrony level of simulated neural signal became closest to that simulated with the actual data. **c** The magnitude of dynamic fluctuations in mean PC (boxplot elements: center line, median; box limits, upper and lower quartiles; whiskers, 1.5 times interquartile range; "plus" symbols, outliers; these definitions are equivalent throughout the paper). Blue shading is used for improving the visibility of the boxplots. The magnitude was quantified by the SD of mean PC across time. The median SD computed from empirical rs-fMRI data is shown by a red vertical line. An asterisk indicates significant differences between the SDs ($p < 0.05$, FDR corrected across all the 10 comparisons). The exact $p$ and $n$ are shown in Supplementary Table 1.

than the magnitude from the constraint-free surrogate data $R$ (Mann–Whitney $U$ test; $p = 3.8 \times 10^{-6}$ and $7.3 \times 10^{-6}$, two-sided, FDR corrected by the Benjamini–Hochberg method; Cliff's delta = 0.55 and 0.53, respectively; see Supplementary Table 1 for full statistical reporting). This result indicates that both geometry and topology of the connectome significantly contributes to dynamic fluctuations between segregation and integration. We confirmed that this finding held true even when mean PC was replaced with mean TPC or modularity. The magnitudes of dynamic fluctuations in mean TPC and modularity obtained from $G$ and $T$ were both greater than the magnitudes from $R$ (see Supplementary Figs. 1 and 2 and Supplementary Tables 2 and 3).

When computing time-resolved functional connectivity in this study, we used a window parameter setting similar to that in Allen et al.[43] as a default (width of windows = 66 TRs [= 47.52 s]; displacement between windows = 3 TRs; see "Methods section" for details). We confirmed that our finding also held true with widths of 44 TRs (= 31.68 s) and 88 TRs (= 63.36 s), as well as displacements of 1 TR and 66 TRs (= the width; see Supplementary Figs. 3–6 and Supplementary Tables 4–7).

**Dynamic fluctuations are not fully accounted for either by geometry, topology, or their combination**. We then examined the extent to which geometry and topology of the connectome can explain the magnitude of dynamic fluctuations generated from the actual connectome data. To this end, we compared the magnitudes of dynamic fluctuations in mean PC between the actual data and the geometry- and/or topology-constrained surrogate data $G$, $T$, and $GT$. We found that the magnitudes of dynamic fluctuations obtained from $G$, $T$, and $GT$ surrogates were smaller than the magnitude from the actual data ($p = 1.4 \times 10^{-12}$, $1.4 \times 10^{-10}$, and $3.9 \times 10^{-7}$, FDR corrected; Cliff's delta = 0.73, 0.66, and 0.52, respectively; see Fig. 4c and Supplementary Table 1). This result suggests that, even with significant contributions from geometry and topology, these global structural features of the connectome alone cannot fully account for dynamic fluctuations between segregation and integration. This finding also held for analyses that examined dynamic fluctuations in mean TPC and modularity. The magnitudes of dynamic fluctuations in mean TPC and modularity from $G$, $T$, and $GT$

were smaller than the magnitudes from the actual data (see Supplementary Figs. 1 and 2 and Supplementary Tables 2 and 3). The magnitudes from $G$, $T$, and $GT$ were also smaller than those from the actual data when the window width was changed to 44 TRs and 88 TRs, or the window displacement was changed to 1 TR and 66 TRs (see Supplementary Figs. 3–6 and Supplementary Tables 4–7).

**Unexplained dynamic fluctuations are shaped by structural connections of the visual network**. In the following, we aim to identify local edge sets in the connectome that were responsible for dynamic fluctuations not explained by a combination of geometry and topology. To approach this aim, we performed simulations using a more constrained version of the surrogate connectome data $GT$. Under the geometric and topological constraints imposed on $GT$, the weight permutation in this new surrogate data were restricted only within an edge set of interest allowing us to examine the contribution of this specific set to the residual dynamic fluctuations. The contribution of a given local edge set was evaluated by comparing the magnitude of dynamic fluctuations simulated using the above surrogate data (referred to as the 'main' surrogate data hereafter) with the magnitude simulated using its control surrogate data. The control surrogate data were also a more constrained version of $GT$, but the weight permutation was restricted within edges not included in the edge set of interest in the main surrogate data. The fraction of permuted edges was equalized between the main and control pair of the surrogate data by randomly selecting edges not to be permuted in the control surrogate data. The local contribution of a specific edge set of interest was assessed by quantifying the decrease in the magnitude of dynamic fluctuations simulated using the main from the control surrogate data.

With this methodology, we examined the local contribution of single RSNs, defined as a cluster of brain regions that coherently fluctuate during the resting state. The RSNs in this study were defined based on a canonical 7-network parcellation[36], which includes the control network (CON), the default mode network (DMN), the limbic system (LIM), the dorsal attention network (DAN), the saliency/ventral attention network (VAN), the somatomotor network (SMN), and the visual network (VIS).

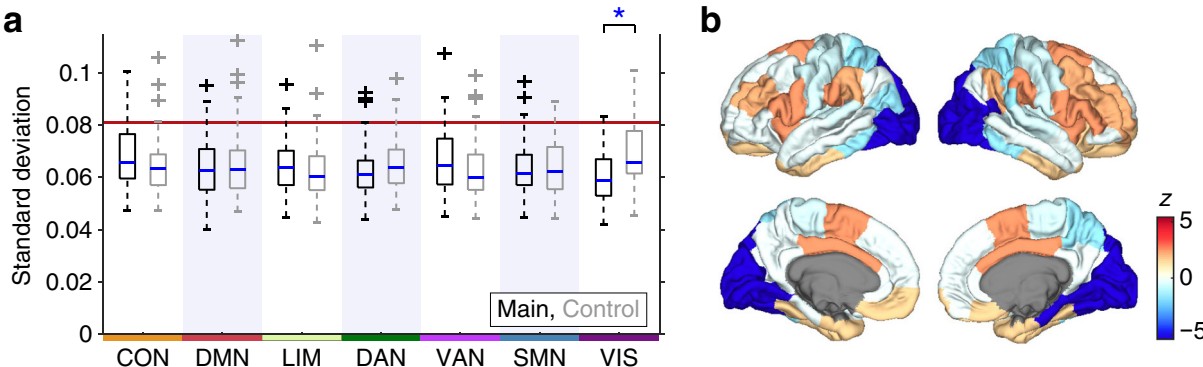

**Fig. 5 The magnitude of dynamic fluctuations in mean PC, obtained from each main–control pair of the RSN-constrained surrogate connectome data. a** The SD of mean PC across time. The left and right boxplots in each RSN column, respectively, are from the main and control RSN-constrained surrogate data. Abbreviations for the RSNs (CON, DMN, LIM, DAN, VAN, SMN, and VIS) are defined in the main text. The median SD computed from empirical rs-fMRI data is shown by a red vertical line. An asterisk indicates significant differences between the SDs ($p < 0.05$, FDR corrected across all the 7 comparisons). The exact $p$ and $n$ are shown in Supplementary Table 8. **b** Cortical surface plots of the $z$-values of differences between the SDs (Main – Control).

We investigated the local contribution of an RSN by specifying the edge set of interest in the main surrogate data as the set of edges structurally connected to at least a node belonging to the corresponding RSN.

Figure 5a shows the magnitudes of dynamic fluctuations in mean PC simulated using each main–control pair of the RSN-constrained surrogate data. We found that the magnitude of dynamic fluctuations significantly decreased in the main surrogate data from its control when the weights of edges connected to the visual network were permuted ($p = 4.0 \times 10^{-7}$, FDR corrected; Cliff's delta = $-0.44$; see also Fig. 5b). In all other cases of RSNs, there was no significant decrease in the magnitude of dynamic fluctuations in the main surrogate data (see Supplementary Table 8). In addition, only in the case of the visual network, the magnitude of dynamic fluctuations was comparable between the main surrogate data and $GT$ ($p = 0.36$) and between the control surrogate data and the actual data ($p = 0.92$), while in all other RSNs these comparisons were with $p < 0.05$ (FDR corrected; see Supplementary Table 8). These findings indicate that structural connections of the visual network disproportionately contribute to shaping the difference between $GT$ and the actual data in the magnitude of simulated dynamic fluctuations, i.e., the residual amount of simulated dynamic fluctuations not explained by geometry or topology of the connectome. The same conclusion was reached for the magnitude of dynamic fluctuations in mean TPC and modularity (see Supplementary Figs. 7 and 8 and Supplementary Tables 9 and 10). Results with window widths of 44 TRs and 88 TRs, or window displacements of 1 TR and 66 TRs, are shown in Supplementary Figs. 9–12 and Supplementary Tables 11–14. Even with these window parameter settings, we observed the greatest contribution of the visual network to the residual dynamic fluctuations.

The large contribution of the visual network to the residual dynamic fluctuations was not due to the differences over the RSNs in the number of permuted edges in the RSN-constrained surrogate data. The fraction of the number of permuted edges averaged over surrogate samples to the number of all non-zero weight edges was 0.022 (CON), 0.131 (DMN), 0.058 (LIM), 0.032 (DAN), 0.058 (VAN), 0.060 (SMN), and 0.078 (VIS).

## Discussion

In this study, we used surrogate connectome data and brain activity simulations to identify which network features of the

connectome are crucial for the emergence of fluctuating functional connectivity patterns between segregation and integration. We addressed this question by quantifying the magnitudes of dynamic fluctuations in mean PC, mean TPC, and modularity while simulating time-resolved rs-fMRI functional connectivity on the surrogate and actual connectome data. We demonstrated that the magnitude of dynamic fluctuations obtained from the geometry- or the topology-constrained surrogate data was greater than the magnitude from randomly permuted surrogate data and that the magnitude from the geometry- and topology-constrained surrogate data was smaller than the magnitude from the actual data. These findings indicate that geometry and topology of the connectome significantly contribute to the appearance of dynamic fluctuations between segregation and integration, but these global network features of the connectome do not fully account for the magnitude of dynamic fluctuations emerged from the actual connectome data. We then evaluated the contribution of structural edges connected to each RSN and found that edges connected to the visual network were most effective in generating the residual dynamic fluctuations not explained by a combination of geometry and topology of the connectome.

The finding that geometry of the connectome did not fully account for the magnitude of dynamic fluctuations between segregation and integration implies that the connectome hosts richer dynamics than expected from its spatial embedding alone. The effect of spatial embedding on the connectome can be seen in the characteristic relationship between weights and lengths of structural connections[33,34], which reflects the principle of wiring cost minimization[31]. Previous studies have shown that wiring cost minimization cannot entirely explain the topology of the connectome and have suggested that the connectome realizes its richer topology at the expense of increasing wiring cost[44,45]. Our study suggests that this also holds true for realization of the fluctuating dynamics of segregation and integration. Increased wiring cost in the connectome may thus serve to not only yield complex network topology, but also enhance dynamic reconfigurations between segregated and integrated functional network organizations in the brain, which are essential for flexibly responding to a variety of cognitive demands[13].

We further demonstrated that topology of the connectome also did not fully explain the magnitude of dynamic fluctuations between segregation and integration, even when geometry was simultaneously taken into account. We examined the contributions from two representative topological features of the connectome, communities[4], and interconnected hubs[7]. The proposed

roles of communities and interconnected hubs in supporting segregation and integration of neural information[11,12] as well as complex dynamics[46] suggest an important role in the emergence of fluctuations between segregation and integration. The significant contribution of topology to dynamic fluctuations shown in "Results section" partially supports this prediction. However, our finding that the geometry- and topology-constrained surrogate data fell short of generating the full extent of dynamic fluctuations obtained from the actual data indicates that additional features other than geometry or topology are involved and motivated us to assess potential additional contributions from local structural edges in the connectome.

Among the seven RSNs[36], we found that structural connections of the visual network were most capable to generate dynamic fluctuations between segregation and integration not accounted for by a combination of geometry and topology. We confirmed that this finding was not due to the variability over the RSNs in the number of permuted edges in the surrogate data and was therefore more likely to result from the specific embedding of the visual network in the overall connectome. One notable feature of the visual network structural connectivity is its high intrinsic density. Numerous studies have demonstrated that nodes in the visual areas indeed maintain dense anatomical interconnections[47]. In our data, we also noted that the density of structural connections within each RSN was the greatest in the visual network (Supplementary Fig. 13). This observation raises the possibility that the locally dense connectivity architecture within the visual network may not only support local processing of visual information, but may also contribute to shaping global network dynamics of functional segregation and integration. The possible contribution to the global dynamics could explain the prominence of the visual network in within-subject variability of rs-fMRI functional connectivity[48], where functional connectivity among regions including the visual areas was shown to be highly variable across a number of scanning sessions in a highly sampled individual brain. The greater variability of visual functional connectivity could result from varying forms of the contribution from the visual areas via their dense structural connections, so that the balance between segregation and integration is appropriately controlled under varying states of the resting brain across scanning sessions.

Although we focused on group-level connectome data in the present study, examining individual differences in the structural organization of the connectome and relating them to empirical dynamics of segregation and integration are important directions for future research. Recent studies established evidence for individual differences in rs-fMRI functional connectivity[49] including its dynamic properties of fluctuating connectivity patterns between segregation and integration[24]. Structural accounts of these individual differences could be identified by assessing their relations to individual differences in the network features of the connectome whose contributions to the segregation–integration dynamics were demonstrated in this study. For instance, individual differences in empirical functional connectivity and its network dynamics could be associated with individual differences in the strength of weight–length relationship, the modularity of network communities, the distinctness of hub-to-hub connections, or the local density of visual structural connectivity. Furthermore, by comprehensively relating them to individual differences in behavioral and cognitive functions, one could explain mechanisms inside the end-to-end relation between reconfigurations of the connectome and declines in behavioral and cognitive performances through e.g. aging[50] or brain diseases[51].

It should be noted that the simulations using the actual connectome data in this study only partially replicate spatial patterns of empirical functional connectivity[30]. A principal reason for this

limitation is that we fitted free parameters in the simulation model to accurately reproduce the temporal dynamics of rs-fMRI functional connectivity in addition to fitting its spatial pattern. In Fukushima and Sporns[30], we showed that parameter fitting optimized only for replicating spatial patterns severely compromised the reproducibility of temporal dynamics and therefore fitted parameters using multiple criteria to balance reproducing both spatial patterns and temporal dynamics. The resulting correlations between simulated and empirical functional connectivity (0.26 for all node pairs; 0.34 for structurally connected node pairs) are comparable to those obtained from simulations using more complex and biophysically realistic models of neural masses or spiking neurons[39]. Future work may be directed at improving the joint reproducibility of spatial patterns and of temporal dynamics. One potential solution is to use more reliable connectome data in the simulations. Replacing undirected connectome data derived from non-invasive tractography with directed connectome data derived from invasive tract tracing[52] could improve the overall replication accuracy of spatial patterns. Another possible solution is to include subcortical areas in the simulations and newly model the role of neuromodulatory systems in actively controlling the neural gain[53]. Given the possible contribution of the neural gain control to the fluctuating dynamics of segregation and integration[14,26], such a model improvement could enhance the reproducibility of temporal dynamics even when parameter fitting is primarily optimized for replicating spatial patterns.

As previously demonstrated in Fukushima and Sporns[30], the magnitude of dynamic fluctuations in global network measures for segregation and integration in the simulations (with the actual connectome data) was smaller than the empirical magnitude (Fig. 4; see also Supplementary Fig. 14 and Supplementary Table 15). One possible reason for this is the lack of physiological noise in simulated rs-fMRI data. Even though no consistent relation was found between empirical fluctuations in mean PC in our dataset and either head movement or respiration[25], residuals of noise in preprocessed empirical rs-fMRI data might partially contribute to the difference between the magnitudes of simulated and empirical fluctuations. Another potential reason for the difference is the lack of subcortical areas in the simulations. Modeling the role of neuromodulatory systems in controlling the neural gain may also enhance the simulated magnitude of dynamic fluctuations between segregation and integration.

In sliding window analysis, we need to determine the width of and the displacement between sliding windows. According to a comprehensive review paper[15] on time-resolved functional connectivity, previous studies suggest that widths of 30–60 s can successfully capture fluctuations in resting-state functional connectivity. Concerning the displacement, both overlapping and non-overlapping sliding windows have been used in the literature. In this study, we used the width of ~45 s and the displacement of ~2 s[43]. We confirmed that the major findings of this study also held true with widths of around 30 s and 60 s, as well as displacements of < 1 s and the window width. These observations suggest that the results are stable across reasonable settings for the width of and the displacement between sliding windows.

In conclusion, we identified specific network features of the connectome that were responsible for the emergence of dynamic fluctuations between segregated and integrated connectivity patterns simulated on the connectome. We found significant contributions to the dynamic fluctuations from geometry and topology of the connectome, as well as distinct local contributions from structural connections of the visual network. These findings provide fine-grained information about how the structure of the connectome promotes ongoing segregation–integration dynamics, thus allowing a deeper understanding of the generative

processes underlying dynamic fluctuations between functional segregation and integration in the brain.

## Methods

**Data acquisition and processing**. Raw structural, functional, and diffusion MRI data were from the WU-Minn Human Connectome Project (HCP)[54]. The HCP recruited the subjects and collected written informed consent, including consent to make de-identified data publicly available. All experimental procedures were approved by the institutional review board at Washington University and all data analyses were performed in accordance with relevant ethical regulations and guidelines of National Institute of Information and Communications Technology. We downloaded the MRI data sets from the public database of the HCP, ConnectomeDB (https://db.humanconnectome.org), and used the data sample labeled as *100 Unrelated Subjects* in this database. We collected this data sample using the user interface of the Connectome DB. From the 100 subjects in this data sample, 15 high motion subjects during rs-fMRI sessions with the criteria used in Xu et al.[55] and one subject categorized as age ≥ 36 were excluded to obtain a quality-controlled data sample from 84 young adults aged between 22 and 35 (44 females). No sample size calculation was performed.

The MRI data in the HCP data sets were acquired with a 32-channel head coil on a modified 3T Siemens Skyra. Image acquisition parameters[56,57] are as follows; structural MRI: 3D MPRAGE T1-weighted sequence, repetition time (TR) = 2,400 ms, echo time (TE) = 2.14 ms, flip angle = 8°, field of view (FOV) = 224 × 224 mm$^2$, 320 slices, and voxel size = 0.7 mm isotropic; rs-fMRI: gradient echo EPI sequence, TR = 720 ms, TE = 33.1 ms, flip angle = 52°, FOV = 208 × 180 mm$^2$, 72 slices, and voxel size = 2 mm isotropic; diffusion MRI: diffusion weighted sequence, TR = 5,520 ms, TE = 89.5 ms, flip angle = 78°, FOV = 210 × 180 mm$^2$, 111 slices, voxel size = 1.25 mm isotropic, three shells of the *b*-value = 1000, 2000, and 3000 s/mm$^2$, and the number of gradient directions = 270 (90 × 3). A notable acquisition parameter is the TR of rs-fMRI data, 720 ms. This TR realized a sampling frequency higher than that in a standard acquisition setting (typically, the TR of rs-fMRI data is 2–3 s) and is suitable for time-resolved functional connectivity analysis. The duration of each run of rs-fMRI data was 14.4 min. The number of runs per subject was four. The rs-fMRI data were acquired with an eyes open condition.

The downloaded MRI data were already preprocessed with the minimal preprocessing pipeline developed by the HCP[58]. In the minimally preprocessed data sets, the MRI data were transformed into Montreal Neurological Institute (MNI) space. Moreover, correction of gradient distortion, motion correction, removal of bias fields, correction of spatial distortion, and normalization of the image intensity were applied to rs-fMRI data; and corrections of susceptibility distortion, eddy current distortion, motion-related artifact, and gradient nonlinearity, as well as normalization of the intensity, were applied to diffusion MRI data. We further preprocessed these minimally preprocessed data using AFNI 16.0.01 (https://afni.nimh.nih.gov), FreeSurfer 5.1.0 (https://surfer.nmr.mgh.harvard.edu), FSL 5.0.9 (https://fsl.fmrib.ox.ac.uk), and custom code implemented in MATLAB R2016a. This includes preprocessing of rs-fMRI data by (i) removing the first 10 s of volumes, (ii) removing outlier volumes and applying interpolation using *3dDespike* in AFNI (on average, 3.6% of the total volumes were interpolated), (iii) regressing out the global, white matter, and cerebrospinal fluid mean time series and the Friston-24 motion time series[59], and (iv) applying detrending and band-pass filtering. The low cut frequency of filtering was specified as 0.021 Hz (1/the width of a sliding window for computing time-resolved functional connectivity) to suppress spurious connectivity fluctuations[60], and the high cut frequency was set to 0.1 Hz.

**Functional connectivity**. The Fisher *z*-transformed Pearson correlation coefficient of node-averaged rs-fMRI signals was used as the weight of functional connectivity between nodes. The node-averaged rs-fMRI signal was obtained by averaging the time courses of voxels within each cortical parcel made from a subdivision of the Desikan-Killiany atlas[61] (the number of parcels: 114), which we downloaded from https://github.com/LTS5/cmp.

Time-resolved functional connectivity was computed using a tapered sliding window[15] with its shape and size specified in a similar manner as in Allen et al.[43]. Specifically, the tapered sliding window was constructed by convolving a rectangle of width = 66 TRs (= 47.52 s) with a Gaussian kernel of σ = 9 TRs (= 6.48 s) and was shifted toward the end of the time series in steps of 3 TRs (= 2.16 s). This procedure yielded 369 sliding windows in total for each run of rs-fMRI data. We constructed time-resolved functional connectivity networks of individual samples as weighted networks without thresholding. We did not threshold the weights here because both positive and negative weights of functional connectivity are useful for identifying the community organization of functional brain networks[4,62].

We examined the robustness of our findings against moderate changes in window parameters. In this robustness analysis, we changed the width of the rectangle window and the σ of the Gaussian kernel for convolution from the default 66 TRs and 9 TRs to either 44 TRs (= 31.68 s) and 6 TRs or 88 TRs (= 63.36 s) and 12 TRs. We also investigated the effect of the displacement between windows by changing it from the default 3 TRs to either 1 TR or 66 TRs (= the width of windows). We used the same preprocessed rs-fMRI data in this robustness analysis,

except when we changed the window width to 44 TRs. In this exception, we filtered out again frequency components < 1/the window width (in this case, 0.032 Hz) to suppress spurious fluctuations[60].

**Structural connectivity**. The structural connectome was constructed using the same parcellation scheme used to calculate functional connectivity. To obtain the structural connectome, distributions of white matter fiber orientations were estimated for each voxel of diffusion MRI data using the generalized *q*-sampling imaging[63]. The estimated orientation distributions were used for reconstructing fiber tracts between cortical parcels by deterministic streamline tractography, as performed in van den Heuvel et al.[64]. The streamline count between a pair of cortical parcels divided by the geometric mean of their surface areas was used as the weight of structural connectivity between nodes.

We employed group-level structural connectivity throughout the paper. The weight of group-level structural connectivity was computed using a consensus thresholding approach that preserves the within- and between-hemisphere streamline length distributions of individual subjects[65]. We adopted the consensus thresholding to exclude structural connections inconsistent with individual subjects from the simulations. The resulting density after the thresholding was 0.19. Non-zero connectivity weights and streamline lengths were averaged over subjects to obtain group-level matrices, which are shown in Fig. 1.

While the weights of structural connectivity were maintained in this study, we employed measures for unweighted networks as well to structural connectivity in the following two cases. First, we used the degree of the structural connectivity network to compute and preserve the weighted rich-club coefficient[7] when constructing topology-constrained surrogate data (see Surrogate connectome data below for more details). Second, we reported the density of structural connections over the whole brain in the above paragraph, and the density of structural connections within each RSN (Supplementary Fig. 13) to discuss the role of the visual network in shaping dynamic fluctuations (see "Discussion section").

**Community detection and modularity**. Communities in connectivity networks were detected by applying a modularity maximization method[66]. In this method, a partition in a network is optimized to maximize the modularity quality function (or simply, modularity) that quantifies the extent to which a network is decomposed into densely connected sub-networks. To deal with negative edge weights in the network of functional connectivity, a generalized version of the modularity quality function[62] $Q^*$ was maximized in this study. $Q^*$ is defined as

$$Q^* = \frac{1}{\nu^+} \sum_{i,j} \left( w_{i,j}^+ - e_{i,j}^+ \right) \delta_{c_i, c_j} - \frac{1}{\nu^+ + \nu^-} \sum_{i,j} \left( w_{i,j}^- - e_{i,j}^- \right) \delta_{c_i, c_j}, \quad (1)$$

where $\delta_{c_i, c_j} = 1$ if nodes $i$ and $j$ are within the same community and $\delta_{c_i, c_j} = 0$ otherwise. The positive and negative superscripts to the edge weight $w_{i,j}$ between nodes $i$ and $j$ are used for separating positive and negative edge weights, where $w_{i,j}^+ = w_{i,j}$ and $w_{i,j}^- = 0$ if $w_{i,j} > 0$, and $w_{i,j}^+ = 0$ and $w_{i,j}^- = -w_{i,j}$ otherwise. The term $e_{i,j}^\pm = s_i^\pm s_j^\pm / \nu^\pm$, where $s_i^\pm = \sum_j w_{i,j}^\pm$ and $\nu^\pm = \sum_{i,j} w_{i,j}^\pm$, is the expected positive or negative edge weight between nodes $i$ and $j$ given a null model preserving each node's strength. When the modularity quality function was computed for a connectivity matrix containing no negative weight, the second term in the right hand side of Eq. (1) was ignored. $Q^*$ was maximized through the Louvain algorithm[67] using the Brain Connectivity Toolbox (https://sites.google.com/site/bctnet), where the resolution parameter γ was specified as the default value one. In each case of modularity maximization for a given connectivity matrix, the Louvain algorithm was applied 100 times with random initial conditions. We regarded the maximum of $Q^*$ over the 100 trials as the maximized modularity score and the partition yielded this score as a final solution of detected communities. The modularity $Q_t^*$ of time-resolved functional connectivity can be used to track dynamic fluctuations between segregated (high $Q_t^*$) and integrated (low $Q_t^*$) connectivity patterns[23,24].

**Participation coefficient and temporal participation coefficient**. The PC is a node-wise measure that quantifies the extent to which a node is connected to other nodes across diverse communities[68]. We computed the PC of time-resolved functional connectivity and its mean averaged over all nodes. Based on communities detected in an instance of time-resolved functional connectivity, the PC of node $i$ at time $t$ was computed as

$$PC_{i,t} = 1 - \sum_{c=1}^{N_{C,t}} \left( \frac{\kappa_{i,c,t}^+}{\kappa_{i,t}^+} \right)^2, \quad (2)$$

where $N_{C,t}$ is the number of detected communities at time $t$, $\kappa_{i,c,t}^+$ is the strength of the positive edge weights of node $i$ at time $t$ within community $c$, and $\kappa_{i,t}^+$ is the strength of the positive edge weights of node $i$ at time $t$ across all communities. The PC was then averaged over all nodes to obtain a network-wise measure $\overline{PC_t}$ (mean PC). The mean PC can also be used as a proxy of dynamic fluctuations between segregated (low $\overline{PC_t}$) and integrated (high $\overline{PC_t}$) patterns of time-resolved functional connectivity[14].

A recent study[42] proposed TPC by extending PC in Eq. (2) to improve its interpretability with time-varying community partitions. The TPC of node $i$ at time

$t$ was calculated using all temporal community partitions as follows:

$$TPC_{i,t} = 1 - \frac{1}{T} \sum_{u=1}^{T} \sum_{c=1}^{N_{c,u}} \left( \frac{\kappa_{i,c,t}^{+}}{\kappa_{i,t}^{+}} \right)^2, \tag{3}$$

where $T$ is the number of sliding windows. Together with modularity and mean PC, we used the TPC averaged over all nodes (mean TPC) to quantify the magnitude of dynamic fluctuations between segregation and integration.

**Surrogate connectome data.** Surrogate connectome data were constructed by permuting non-zero structural connectivity weights of the actual connectome data. During the weight permutation, geometric and/or topological features of the actual connectome were preserved by constraints to assess their contributions to the emergence of dynamic fluctuations between segregation and integration. After the weight permutation, weights of the surrogate connectome data were adjusted to preserve the strength sequence of the actual data[33,34] using *strengthcorrect.m* in https://github.com/breakspear/geomsurr. We referred to the strength-preserved surrogate data without constraint as $R$ (= $R_{ss}$ in Gollo et al.[34]). Procedures to construct the surrogate data with geometric, topological, and both of these constraints $G$ (= $G_{ss}$ in Gollo et al.[34]), $T$, and $GT$, as well as the $GT$-based RSN-constrained surrogate data, were described in detail below.

*Geometry-constrained surrogate data.* A geometric constraint was imposed on the surrogate data $G$ and $GT$ to preserve the spatial relationship between weights and lengths of structural connections[33,34]. In these geometry-constrained surrogate data, the weight permutation was restricted within each equal-width bin of lengths[34] (Fig. 2a). The number of bins needs to be large enough to capture the weight–length relationship in the actual data, while it must not be very large to secure variations in surrogate samples. We specified the number of bins in this study as 30. We obtained this number by decreasing the number of bins from 100 in steps of 10 (100, 90, 80, and so on) and then finding the case when the percentage of bins containing only three or fewer edges became ≤ 10% for the first time. We confirmed that all bins contained more than one edge with this bin number setting. The adjustment of weights in $G$ and $GT$ to preserve the strength sequence did not essentially alter the weight–length relationship of the actual connectome data (see Fig. 3b).

*Topology-constrained surrogate data.* Two topological constraints were imposed on the surrogate data $T$ and $GT$ to preserve communities and interconnected hubs of the network of structural connections. In these topology-constrained surrogate data, the weight permutation was restricted within each community detected in the actual connectome or within edges connecting whatever pairs of two different communities (Fig. 2b), and simultaneously within each edge category consisting of edges connecting a node of a certain degree $d$ and another node of degree ≥ $d$ (Fig. 2c). Permuting weights within each of such degree-based edge categories preserves the weighted rich-club coefficient[7], i.e. the ratio of the sum of edge weights between hub nodes to the sum of all edge weights, over different degree thresholds to define hub nodes. The adjustment of weights in $GT$ did not influence on the extent to which the surrogate data preserved communities and interconnected hubs of the actual data (see Fig. 3c, d). The weight adjustment in $T$ also did not essentially change the extent to which the community structure was preserved (Fig. 3c), whereas it made the interconnected hub structure less pronounced as the weighted rich-club coefficient of $T$ became closer to that of $R$ (Fig. 3d). Therefore, the contribution of topology to dynamic fluctuations as assessed by $T$ in "Results section" was mainly represented by the contribution from the community structure alone.

*RSN-constrained surrogate data for assessing local contributions of RSNs.* Surrogate data to examine the contribution of local edge sets in the connectome were constructed under the geometric and topological constraints imposed on $GT$. This approach was chosen to uncover which local edge sets were responsible for dynamic fluctuations not accounted for by a combination of geometry and topology of the connectome. We searched for them in a spatial resolution of RSNs defined based on the 7-network parcellation[36], where each cortical parcel (node) in this study was assigned to one of the following seven RSNs, the control network, the default mode network, the limbic system, the dorsal attention network, the saliency/ventral attention network, the somatomotor network, and the visual network, by selecting an RSN of the maximum area of cortical surface overlap. The surrogate data were constructed as a more constrained version of $GT$, in which permuting weights was allowed only within edges connected to at least a node in an RSN of interest. We quantified the effect of permuting edges connected to an RSN of interest on the segregation–integration dynamics by comparing the magnitude of dynamic fluctuations generated from the above main surrogate data and that from another more constrained version of $GT$ referred to as the control surrogate data, where permuting weights was performed only within edges not connected to the RSN of interest. The number of edges that can be permuted in the control surrogate data was always greater than that in the main surrogate data when there was no correction. To properly compare the magnitudes of dynamic fluctuations between each main–control pair of the surrogate data, the number of permuted edges in the control surrogate data was reduced by permuting weights of only a

randomly selected subset of edges. By optimizing the size of this subset, the difference in the fraction of the number of permuted edges to the number of all non-zero weight edges between each main–control pair of the surrogate data became < $10^{-4}$ on average for each RSN.

**Simulation of resting-state activity.** A variant of the Kuramoto model[37], in which phase oscillators are coupled based on the structural connectome, was used for simulating spontaneous oscillatory neural signal. The Kuramoto model is a simple dynamic model while it can generate synchronization dynamics of neural populations[69] and can also reproduce a variety of empirical findings about resting-state functional connectivity[38,39] including its network dynamics[30]. In this model, the periodic dynamics of a phase oscillator at node $i$ is expressed by its phase $\theta_i(t)$, which obeying the following differential equation:

$$\frac{d\theta_i}{dt} = 2\pi f + k \sum_{j=1}^{N} c_{i,j} \sin\left( \theta_j(t - \tau_{i,j}) - \theta_i(t) \right), \tag{4}$$

where $f$ is the natural frequency, set to 60 Hz in the gamma band for all nodes[39], $N$ is the number of nodes, $k$ is the global coupling constant, $c_{i,j}$ is the weight of structural connectivity between nodes $i$ and $j$, and $\tau_{i,j}$ is the delay of interactions between nodes $i$ and $j$. The weight $c_{i,j}$ was normalized so that the mean non-zero edge weights of structural connectivity equals one, and $k$ controlled the overall coupling strength. The delay $\tau_{i,j}$ was simply assumed to be $L_{i,j}/v$ as in previous studies[29], where $L_{i,j}$ is the streamline length between nodes $i$ and $j$, and $v$ denotes the conduction velocity. Equation (4) was solved numerically using the Heun method with the step size of 0.2 ms. The initial value of the phase was set at random from the range $[0, 2\pi]$ and the initial history of the phase was obtained by running the simulations without interactions for a short duration. Transient dynamics were removed by discarding the initial 20 s of simulation data.

Simulated rs-fMRI signal was derived from the simulated phase time series of coupled oscillators. The simulated phase $\theta_i(t)$ was first transformed into the amplitude space as $r_i(t) = \sin(\theta_i(t))$ to obtain the amplitude of simulated oscillatory neural signal. Then, after downsampled to the sampling frequency of 1 kHz, the simulated neural signal was converted to simulated rs-fMRI signal using the Balloon/Windkessel hemodynamic model[40,41]. The simulated rs-fMRI signal was band-pass filtered using the same filter as applied to empirical rs-fMRI signal. In addition, the simulated rs-fMRI signal was downsampled to match the sampling frequency between the simulated and empirical data, and the global mean signal was also regressed out for consistency. The number of time points in a single simulation sample was the same as the number of time points in a single run of the empirical data. Time-resolved functional connectivity and its global network measures (mean PC, mean TPC, and modularity) were also computed for the simulated data in the same manners to track dynamic fluctuations between segregation and integration.

The global synchrony level of coupled oscillators in the simulation model was evaluated through the order parameter $r(t)$:

$$r(t)e^{i\psi(t)} = \frac{1}{N} \sum_{n=1}^{N} e^{i\theta_n(t)}, \tag{5}$$

where $\psi(t)$ is the phase of the global ensemble of oscillators. The order parameter $r(t)$ varies between 0 and 1, quantifying the uniformity of phases over all oscillators. The average of $r(t)$ over time was used for quantifying the level of global synchrony of the whole coupled oscillator system[70].

**Model parameter search.** The simulation model in Eq. (4) has two free parameters, the global coupling constant $k$ and the mean delay $\bar{\tau} = \bar{L}/v$, where $\bar{L}$ is the streamline length averaged over all edges of non-zero structural connectivity weights. In Fukushima and Sporns[30], these model parameters were optimized so that simulated rs-fMRI data generated from the structural connectome can reproduce empirical properties of functional connectivity measured over the entire scan durations, as well as those of time-resolved functional connectivity including dynamic fluctuations in its global network measures. The comparison between the simulated and empirical data was performed in a two-stage manner. At the first stage, parameter sets yielding the Pearson correlation coefficient of long-time scale functional connectivity at structurally connected edges > 0.33 and the Kolmogorov–Smirnov distance of the edge weight distributions of time-resolved functional connectivity < 0.33 were extracted from 2.5 to 70 (step size: 2.5) for $k$ and 2–17 ms (step size: 1 ms) for $\bar{\tau}$. From these extracted parameter sets, the one that best reproduced the magnitudes of dynamic fluctuations in mean PC and modularity was identified at the second stage. After this comparison procedure, optimal values for reproducing the empirical properties were found to be $k = 55$ and $\bar{\tau} = 12$ ms for the data that were also used in the present study as the actual connectome data. The selected $\bar{\tau}$ corresponds to $v = 7.0$ m/s, which is in a realistic range of the conduction velocity of the primate brain (5–20 m/s)[71]. In all, 100 simulation samples with this parameter set were generated in Fukushima and Sporns[30] and were employed as the simulation samples obtained from the actual connectome data in "Results section".

Since spatial connectivity patterns in the surrogate connectome data were changed by the weight permutation, the goodness of fit of empirical connectivity profiles cannot be used for specifying the free parameters in the simulations with

the surrogate data. Instead, we employed the global synchrony level (the temporal average of $r(t)$ in Eq. (5)) of simulated neural signal as a criterion for the model parameter search. Selecting the free parameters that make the global synchrony level comparable between simulation samples generated from the surrogate and actual connectome data allows meaningfully comparing the magnitudes of dynamic fluctuations in mean PC, mean TPC, and modularity between the surrogate and actual data.

As this parameter search needed to be performed for each sample of the surrogate connectome data, we reduced the total amount of computation by fixing the mean delay $\bar{\tau}$ to 12 ms (the same value as in the simulations with the actual data), which yielded a realistic conduction velocity as already mentioned above. The global coupling constant $k$ was changed from 2.5 to 70 in steps of 2.5. The simulations were run with these values of $k$ for each surrogate connectome sample, and the global synchrony level of simulated neural signal was computed for each $k$. Then, the difference between the synchrony level from the surrogate data and the synchrony level (averaged over all simulation samples) from the actual data was evaluated. The $k$ in the surrogate data that yielded the synchrony level most closely approximated that from the actual data was specified as a value from which the above difference became the minimum over the candidates of $k$. When this minimum difference was greater than the radius (i.e. the maximum difference from the mean) of the distribution of the synchrony level from the actual data (this was 0.085), then the corresponding surrogate connectome sample was discarded and a new surrogate sample was regenerated. Finally, we obtained 50 synchrony-level controlled simulation samples from each type of the surrogate connectome data $R$, $G$, $T$, and $GT$.

We found that the $k$ selected for the surrogate data $GT$ was within a range between 50 and 60 (see Fig. 4b). We took advantage of this result to reduce the parameter search space of $k$ for the RSN-constrained surrogate data, which were with the geometric and topological constraints imposed on $GT$. For these surrogate data, $k$ was changed from 50 to 60 in steps of 2.5 during the parameter search. With this reduced parameter search space, we generated $100 \times 2$ simulation samples from each main and control sets of the RSN-constrained surrogate data for each RSN.

**Reporting summary**. Further information on research design is available in the Nature Research Reporting Summary linked to this article.

## Data availability
The MRI data used in this study are available from the data sample *100 Unrelated Subjects* in the HCP's public database, ConnectomeDB (https://db.humanconnectome.org).

## Code availability
Custom code to construct all types of the surrogate connectome data employed in this study is provided as Supplementary Software in Zenodo[72] (we used MATLAB r2016a to run this code). Custom code that is not deemed central to the conclusions is also available from the corresponding author upon request.

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

## Acknowledgements

We thank Marcel A. de Reus and Martijn P. van den Heuvel for processing diffusion MRI data. This work was supported by JSPS Research Fellowship for Young Scientists (PD 201800004), JSPS KAKENHI Grants (JP18J00004 and JP19K16887), JNNS 30th Anniversary Research Grant, and NAIST Foundation Education and Research Activities Grant to M.F.; and NIH R01 AT009036-01 to O.S. Data were provided by the Human Connectome Project, WU-Minn Consortium (Principal Investigators: David Van Essen and Kamil Ugurbil; 1U54MH091657) funded by the 16 NIH Institutes and Centers that support the NIH Blueprint for Neuroscience Research; and by the McDonnell Center for Systems Neuroscience at Washington University.

## Author contributions

M.F. and O.S. designed the study, interpreted the results, and wrote the paper. M.F. performed the analyses.

## Competing interests

The authors declare no competing interests.
