## [Peer Review File · Communications Biology]

Reviewers' comments:

Reviewer #1 (Remarks to the Author):

In the article "Structural determinants of dynamic fluctuations between segregation and integration on the human connectome", the authors used surrogate connectomes and functional connectivity simulations to explore the crucial brain network features to explain the dynamic fluctuations between segregation and integration in resting-state functional connectivity.

This article shows that both topology and geometry of the connectome explains a significant part of the functional connectivity fluctuations when compared to a randomly permuted surrogate connectome. Additionally, the authors explored what measure contributed to the remaining fluctuations not accounted for by the geometry and topology. By constraining the edges permuted to specific resting-state functional networks, they showed that the connections of the visual network could explain the residual fluctuations.

While this work is quite theoretical, the methods and their combination will undoubtedly be valuable to other researchers. Besides, the insights on the network properties of the connectome related to the functional connectivity dynamic are a significant advance for the understanding of functional and structural connections relationship.

Main comments:

I think the fluctuations between segregation and integration would deserve some clarification as it is the core of the findings. Because throughout the paper, the fluctuations are merely the measure we compare in the different models, but the introduction could develop a bit more on the real-life (as far as we can call fMRI real-life of course) aspects of this concept. I guess just a paragraph would suffice.

In the Results, the fraction of permuted edges for GT is less than half on average over the surrogate sample. Does it mean that there was only half of the edge permuted in the comparisons between the GT surrogates and the actual connectome? I am not sure to grasp how it would influence the result or not.

Minor comments:

Methods:

"The number of bins in this study was specified as 30. This number was obtained as a value from which the percentage of the number of edges in a bin ≤ 3 became $\leq 10\%$ across bins for the first time when the number of bins was decreased from 100 in steps of 10."

I am having a hard time understanding that bit, would it be possible to clarify that a bit?

Reviewer #2 (Remarks to the Author):

This is a clear and well-written study, aiming to identify anatomical brain network signatures of dynamic segregation/integration switch in functional brain connectivity.

Despite the fact that simulated data can only partially reproduce real ones, I find the work interesting with many original points. I have a number of comments that will hopefully help the authors improving their manuscript.

1. My main concern is the stability of the results across different time scales. Only one setting has been tested for the sliding time window when computing FC, so we don't know to what extent results may change across different time scales (both in terms of time window length and amount of displacement). This information should be computed, reported and discussed.

2. My second concern is the use of the ratio of as a measure of dynamic fluctuation. Why not simply reporting the SD of PC for simulated and real network separately? This will perhaps make the interpretation easier, too.

3. Somewhat related to the previous point, why the mean PC for real data is higher than simulated one ($SD < 1$)? Can it just be that real fMRI data are noisier (variable) than simulated ones? Perhaps authors do have an explanation but this should be better explicated.

4. A minor point that should be clarified is how brain networks have been filtered. For example, for FC networks it is not explicitly said if they have considered the fully weighted connected networks or not. This should be motivated either way. Why not for example using the same approach they used for filtering anatomical brain networks? Also weights seem to be maintained in general, but sometimes I see node degree, which is for unweighted networks (eg, Fig 3c). This should be better explained and presented to avoid confusion.

5. If networks have been filtered, it would be interesting to report the resulting connection density, too.

6. Introduction. When referring to network cores, authors might be interested to consider a recent work on core-periphery structure in the human connectome integrating topological info from both structural and functional networks¹.

7. Finally, I'm not totally clear if the results in the last section are for real or simulated data. Please clarify.

References

1. Battiston, F., Guillon, J., Chavez, M., Latora, V. & De Vico Fallani, F. Multiplex core-periphery organization of the human connectome. *J. R. Soc. Interface* 15, 20180514 (2018).

Structural determinants of dynamic fluctuations between segregation and integration on the human connectome

Makoto Fukushima, Olaf Sporns

August 4, 2020

We deeply appreciate valuable comments and feedback from the reviewers to our manuscript. We would like to also thank the editors for giving us an opportunity to revise our manuscript. In this revision, we have addressed all concerns raised by the reviewers. We reply to the reviewers' specific comments one by one in this response letter by pointing out how we have addressed them in the revised manuscript. Below we present comments from the reviewers in *italic* and our responses in roman font. We hope that our revised manuscript and response satisfactorily address all comments from the reviewers.

1 Response to reviewers' comments

1.1 Reviewer 1 (4 responses)

1) *While this work is quite theoretical, the methods and their combination will undoubtedly be valuable to other researchers. Besides, the insights on the network properties of the connectome related to the functional connectivity dynamic are a significant advance for the understanding of functional and structural connections relationship.*

Response: We are grateful to the reviewer for the positive and encouraging comments to our manuscript.

2) *Main comments:*

I think the fluctuations between segregation and integration would deserve come clarification as it is the core of the findings. Because throughout the paper, the fluctuations are merely the measure we compare in the different models, but the introduction could develop a bit more on the real-life (as far as we can call fMRI real-life of course) aspects of this concept. I guess just a paragraph would suffice.

Response: We would like to thank the reviewer for this thoughtful suggestion. By extending the second paragraph of **Introduction**, we have added explanations about fluctuations between segregation and integration. In this extended paragraph in the revised manuscript, we describe the basic concept of dynamic changes between segregation and integration first and then introduce their relations to human cognition and behavior referring to recent fMRI studies.

3) *In the Results, the fraction of permuted edges for GT is less than half on average over the surrogate sample. Does it mean that there was only half of the edge permuted in the comparisons between the GT surrogates and the actual connectome? I am not sure to grasp how it would influence the result or not.*

Response: Yes, due to the constraints imposed on geometry and topology, only 40% on average of the edges of non-zeros weights were permuted in the surrogate data *GT* compared to the actual connectome. As the lower fraction of the number of permuted edges enhances the degree to which geometry and topology are preserved in surrogate data, one can speculate that the lower fraction itself indirectly increases the magnitude of dynamic fluctuations in global network

measures for segregation and integration. However, this was not always the case with our data. For instance, the fractions of the number of permuted edges in G and GT were 0.98 and 0.40, respectively, but the magnitudes of dynamic fluctuations in modularity with G and GT were comparable to each other (see Supplementary Fig. 2). In contrast, the fractions of the number of permuted edges in R and G were comparable, but the magnitudes of dynamic fluctuations in modularity were significantly higher with G . These results suggest that the constraints imposed on surrogate data, rather than the fraction of the number of permuted edges, were more critical to the differences observed in the magnitude of dynamic fluctuations among the surrogate sets.

4) *Minor comments:*

Methods:

"The number of bins in this study was specified as 30. This number was obtained as a value from which the percentage of the number of edges in a bin ≤ 3 became $\leq 10\%$ across bins for the first time when the number of bins was decreased from 100 in steps of 10."

I am having a hard time understanding that bit, would it be possible to clarify that a bit?

Response: We thank the reviewer for helping us to clarify the text that was hard to read. We have corrected this text in the revised manuscript as follows: "We specified the number of bins in this study as 30. We obtained this number by decreasing the number of bins from 100 in steps of 10 (100, 90, 80, and so on) and then finding the case when the percentage of bins containing only three or fewer edges became $\leq 10\%$ for the first time."

1.2 Reviewer 2 (8 responses)

1) *This is a clear and well-written study, aiming to identify anatomical brain network signatures of dynamic segregation/integration switch in functional brain connectivity.*

Despite the fact that simulated data can only partially reproduce real ones, I find the work interesting with many original points. I have a number of comments that will hopefully help the authors improving their manuscript.

Response: We would like to thank the reviewer for giving us the positive comments and constructive feedback to improve the quality of the previous version of our manuscript.

2) 1. *My main concern is the stability of the results across different time scales. Only one setting has been tested for the sliding time window when computing FC, so we don't know to what extent results may change across different time scales (both in terms of time window length and amount of displacement). This information should be computed, reported and discussed.*

Response: The reviewer pointed out an important issue and we agree with the reviewer's suggestion. In accordance with the reviewer's comment, we performed robustness analyses using different window widths and displacements. According to Preti et al. (2017), previous studies suggest that the width of 30–60 s can successfully capture fluctuations in resting-state functional connectivity. Concerning the displacement, both overlapping and non-overlapping sliding windows have been used in the literature. In this study, we used a window parameter setting similar to that in Allen et al. (2014) (width = 66 TRs [= 47.52 s]; displacement = 3 TRs) as a default. We have confirmed that the major findings of this study also held true with the widths of 44 TRs (= 31.68 s) and 88 TRs (= 63.36 s), as well as the displacements of 1 TR and 66 TRs (= the width). These observations suggest that the results are stable across reasonable settings for the width of and the displacement between sliding windows.

We have added descriptions about results, discussion, and methods of these robustness analyses to the corresponding sections of the revised manuscript. Figures and tables of them are

presented in Supplementary Figs. 3–6 and 9–12 and Supplementary Tables 4–7 and 11–14.

3) 2. *My second concern is the use of the ratio of as a measure of dynamic fluctuation. Why not simply reporting the SD of PC for simulated and real network separately? This will perhaps make the interpretation easier, too.*

Response: We thank the reviewer for raising this issue, which allows us to improve the presentation of results. Following the reviewer’s suggestion, we replaced the ratio of simulated to empirical SD with the simulated SD itself. We now show the median of the empirical SD across all subjects and runs using a red vertical line in Figs. 4 and 5 and Supplementary Figs. 1–12. Furthermore, we also present distributions of the empirical SD in Supplementary Fig. 14 for direct comparisons between the magnitudes of simulated and empirical SDs.

4) 3. *Somewhat related to the previous point, why the mean PC for real data is higher than simulated one ($SD < 1$)? Can it just be that real fMRI data are noisier (variable) than simulated ones? Perhaps authors do have an explanation but this should be better explicated.*

Response: We are grateful that the reviewer has brought this important point to our attention. As suggested by the reviewer, one possible reason for this is the lack of physiological noise in simulated rs-fMRI data. Even though no consistent relation was found between empirical fluctuations in mean PC in our dataset and either head movement or respiration (Fukushima et al., 2018, Brain Struct. Funct.), residuals of noise in preprocessed empirical rs-fMRI data might partially contribute to the difference between the magnitudes of simulated and empirical fluctuations. Another potential reason for the difference is the lack of subcortical areas in the simulations. Modeling the role of neuromodulatory systems in controlling the neural gain may also enhance the simulated magnitude of dynamic fluctuations between segregation and integration. We have included these explanations in the **Discussion** of the revised manuscript.

5) 4. *A minor point that should be clarified is how brain networks have been filtered. For example, for FC networks it is not explicitly said if they have considered the fully weighted connected networks or not. This should be motivated either way. Why not for example using the same approach they used for filtering anatomical brain networks? Also weights seem to be maintained in general, but sometimes I see node degree, which is for unweighted networks (eg, Fig 3c). This should be better explained and presented to avoid confusion.*

Response: We apologize for leaving it unclear whether networks were filtered or not in the previous version of our manuscript. We have clarified this in the revised manuscript by adding explanations to the sections **Functional connectivity** and **Structural connectivity** in **Methods**.

The reasons why we chose two different approaches regarding the filtering (i.e. thresholding) are a) that functional connectivity has negative weights but structural connectivity does not and b) that we focused on time-resolved functional connectivity at individual-level data but structural connectivity focused on group-level data. In this study, we constructed time-resolved functional connectivity networks of individual samples as weighted networks without thresholding. We did not threshold the weights here because both positive and negative weights of functional connectivity are useful for identifying the community organization of functional brain networks (Rubinov and Sporns, 2011). The group-level structural connectivity network was primarily used for simulating resting-state activity in this study. For this network, we adopted a consensus thresholding method (Betz et al., 2019) to exclude structural connections inconsistent over individual subjects from the simulations.

We have also added a paragraph to **Structural connectivity** in **Methods** to explain the cases when we employed measures typically used for unweighted networks to the weighted structural connectivity network. First, we used the degree of the structural connectivity network to compute and preserve the weighted rich-club coefficient (van den Heuvel and Sporns, 2011) when constructing topology-constrained surrogate data. This is the reason why the degree appears in Fig. 3c. The weighted rich-club coefficient is an extended version of the rich-club coefficient for weighted networks, but is usually defined as a function of the degree. Second, in **Discussion**, we mentioned the density of structural connections within each RSN (Supplementary Fig. 13) to discuss the role of the visual network in shaping dynamic fluctuations. We now also report the density of structural connections over the whole brain in **Structural connectivity** in **Methods** to address the reviewer’s comment below.

6) 5. *If networks have been filtered, it would be interesting to report the resulting connection density, too.*

Response: As explained in our response above, functional connectivity networks were not filtered. The resulting density of the structural connectivity network after the filtering was 0.19. We have added this information to **Structural connectivity** in **Methods**.

7) 6. *Introduction. When referring to network cores, authors might be interested to consider a recent work on core-periphery structure in the human connectome integrating topological info from both structural and functional networks1.*

References

1. Battiston, F., Guillon, J., Chavez, M., Latora, V. & De Vico Fallani, F. *Multiplex core-periphery organization of the human connectome. J. R. Soc. Interface* 15, 20180514 (2018).

Response: We thank the reviewer for pointing out this important paper. We have cited this in **Introduction** of the revised manuscript as a reference of densely interconnected network cores in the brain.

8) 7. *Finally, I’m not totally clear if the results in the last section are for real or simulated data. Please clarify.*

Response: We apologize for ambiguous descriptions in the previous version of the manuscript. These results are for simulated data. We have made this clear by modifying text in the first and third paragraph of the last section of **Results** in the revised manuscript.

REVIEWERS' COMMENTS:

Reviewer #1 (Remarks to the Author):

The authors have answered all of my questions and remarks. I think the paper can be published.

Reviewer #2 (Remarks to the Author):

Authors have satisfactorily addressed all my concerns.